# Structural Design and Experimental Studies of Resonant Fiber Optic Scanner Driven by Co-Fired Multilayer Piezoelectric Ceramics

**DOI:** 10.3390/mi14030517

**Published:** 2023-02-23

**Authors:** Liyuan He, Zhiyi Wen, Boquan Wang, Xiaoniu Li, Dawei Wu

**Affiliations:** State Key Laboratory of Mechanics and Control of Mechanical Structures, Nanjing University of Aeronautics and Astronautics, Nanjing 210016, China

**Keywords:** piezoelectric actuation, co-fired multilayer piezoelectric ceramics, fiber optic scanner, dynamics characters, finite element method

## Abstract

Piezo-driven resonant fiber optic scanners are gaining more and more attention due to their simple structure, weak electromagnetic radiation, and non-friction loss. Conventional piezo-driven resonant fiber optic scanners typically use quadrature piezoelectric tubes (piezo tubes) operating in 31-mode with high drive voltage and low excitation efficiency. In order to solve the abovementioned problem, a resonant fiber scanner driven by co-fired multilayer piezoelectric ceramics (CMPCs) is proposed in which four CMPCs drive a cantilevered fiber optic in the first-order bending mode to achieve efficient and fast space-filling scanning. In this paper, the cantilever beam vibration model with base displacement excitation was derived to provide a theoretical basis for the design of the fiber optic scanner. The finite element method was used to guide the dynamic design of the scanner. Finally, the dynamics characteristics and scanning trajectory of the prepared scanner prototype were tested and compared with the theoretical and simulation calculation results. Experimental results showed that the scanner can achieve three types of space-filling scanning: spiral, Lissajous, and propeller. Compared with the structure using piezo tubes, the designed scanner achieved the same scanning range with smaller axial dimensions, lower drive voltage, and higher efficiency. The scanner can achieve a free end displacement of 10 mm in both horizontal and vertical directions under a sinusoidal excitation signal of 50 V_p-p_ and 200 Hz. The theoretical, simulation and experimental results validate the feasibility of the proposed scanner structure and provide new ideas for the design of resonant fiber optic scanners.

## 1. Introduction

The application of optical scanning systems takes on critical significance in optical precision engineering, industrial flaw detection, displacement sensing, medical imaging and other fields. In these fields, there is an urgent need for fast, high-precision and wide field-of-view (FOV) scanning. In addition, industrial flaw detection and medical imaging fields have proposed more rigorous requirements for miniaturization, low voltage, and low power consumption of the scanners, i.e., the core components of the optical scanning systems. According to the driving method, optical scanners are primarily classified into mechanical, galvanometric, electronic, micro-electro–mechanical systems (MEMS), piezoelectric systems, etc. [1,2,3,4,5]. Currently, the most used mechanical optical scanners cannot easily meet the requirements of large angles, low errors, or high-speed scanning. Other types of scanners also have some limitations. For instance, galvanometer optical scanners are bulky and difficult to miniaturize. Voice coil motor-driven scanners are capable of achieving large angles and high-linearity scanning, whereas they scan at low speeds. MEMS devices are difficult to be widely used for their complex manufacturing process, high manufacturing cost and high price [6,7,8]. The piezoelectric scanners have advantages in terms of their simple structure, weak electromagnetic radiation, and no friction loss, and are capable of obtaining relatively high scanning linearity and deflection accuracy after correction [9,10,11]. Thus, they can accommodate an increasingly wide range of applications [12,13,14].

The tubular piezoelectric actuator, also known as piezoelectric tube (piezo tube), has the advantages of a simple structure, fast response, low power consumption and stable and continuous operation. It has been extensively employed as the probe driving actuator and applied to three-dimensional (3D) displacement systems for scanning probe microscope systems [15]. In general, such scanners operate under non-resonant conditions and exhibit positioning accuracy at the nanometer level, whereas it is difficult to obtain a large range of scans. This limitation can be overcome by using resonance scanning technology. The scanning fiber endoscopy (SFE) investigated by the University of Washington in the United States utilizes cantilever fiber optic driven by a quadrature piezo tube for two-dimensional (2D) transverse resonance scanning under the first-order transverse (bending) vibration mode. The SFE technique is capable of obtaining optical images with high frame rates [16,17]. Besides standard endoscopic optical imaging, the research team successfully demonstrated the feasibility of SFE in more complex optical imaging techniques (e.g., wide-field fluorescence, photometric stereoscopic, confocal, two-photon fluorescence and other optical imaging techniques) [18,19]. Wang Ying and Li Zhi replaced the circular piezo tube by combining four piezoelectric ceramic pieces into a square piezo tube to drive the cantilever fiber optic to achieve optical imaging of the sample [20]. Compared with the circular piezo tube, the square piezo tube is less costly and easier to process, but it also has the problem of insufficient driving force. In addition to the piezo tube, a miniaturized optical scanner can be constructed by bonding together two piezoelectric bimorphs with bending axes perpendicular to each other. However, the asymmetry of the bimorph structure causes the dynamics model to be more complex [21]. Existing research has suggested that the piezoelectric scanner is characterized by a compact structure, simple driving method, as well as stable and continuous imaging. However, the driving voltages of the piezo tubes and the piezoelectric bimorphs are relatively high (usually up to hundreds of volts), and the operation is conducted based on the 31-mode, such that the excitation efficiency is low. The low excitation efficiency remains though the driving voltage can be reduced by changing the structure. The possible cause of the abovementioned problem is the low piezoelectric coefficient of the piezoelectric material. Accordingly, the choice of piezoelectric materials exhibiting high piezoelectric coefficients and the use of higher-efficiency drive modes can reduce the loss of drive performance as the drive voltage decreases, thereby increasing drive efficiency, improving safety and expanding the application range of the devices.

Existing research has suggested that co-fired multilayer piezoelectric ceramic (CMPC) effectively reduces the driving voltage of piezoelectric actuators with little impact on the drive performance [22]. This is due to the fact that the *d_33_* coefficient of CMPCs is tens of thousands while that of Lead zirconate titanate (PZT) is several hundred. In other words, CMPCs can produce the same displacement as PZTs at a lower driving voltage and a higher power density. [23]. For example, New Scale Technologies developed a miniature piezoelectric actuator consisting of four CMPCs arranged lengthwise along a pipe, a threaded nut and a matching screw [24]. Its overall size is only 1.82 mm × 1.82 mm × 6 mm and it can work under 2.3 V_p-p_. Yoseph Bar-Cohen et al. proposed a rotating piezoelectric actuator comprising 12 CMPCs, which is capable of operating at a lower driving voltage and exhibits a higher power density [25]. In addition, Wen Zhiyi et al. developed an in-plane traveling wave rotating piezoelectric actuator driven by CMPCs, which can operate smoothly at a driving voltage of 5 V_p-p_ [26]. Therefore, CMPCs can replace piezo tubes or piezoelectric bimorphs in conventional resonant fiber optic scanners to achieve the miniaturization and low-voltage drive requirements of the optical scanners.

In this work, a resonant fiber optic scanner driven by four CMPCs was proposed to solve the problems of high driving voltage and the low excitation efficiency of traditional piezoelectric optical scanners. The dynamics model of the scanner was established. Moreover, the finite element simulation analysis verified the correctness of the dynamics model and the feasibility of different scanning patterns. Lastly, the accuracy of the dynamics model and simulation results was verified through experimental comparison. The results suggest that the proposed fiber optic scanner is characterized by a simple structure, weak electromagnetic radiation, and no friction loss while producing a larger scanning range at a lower driving voltage. Furthermore, compared with the piezo tube-driven scanner, the proposed fiber optic scanner exhibits a shorter axial size and a more compact overall structure, which is expected to serve as a vital general device in optical precision engineering, industrial flaw detection, displacement sensing, medical imaging, etc.

## 2. Structure Design and Working Principle

### 2.1. Structure Design

The structure of the resonant fiber optic scanner designed is shown in Figure 1a. The scanner is mainly composed of a shell, a base, four CMPCs, a connector, a cantilever capillary tube, and a single-mode optical fiber. Among them, the shell is used to protect the core scanning components and to hold the base. There is a through-hole in the center of the base for passing through the optical fiber and the electrode wires of the CMPCs. Four CMPCs are symmetrically bonded to the base, as shown in Figure 1b, for exciting the horizontal and vertical transverse vibration modes of the capillary tube, respectively. In addition, the other end of the four CMPCs is bonded to the connector. The connector is a conical structure with a through-hole in the center to magnify the displacement and to connect and fix the capillary tube. Finally, the optical fiber passes through the base, connector and capillary tube, and scans as the capillary tube vibrates.

### 2.2. Driving Principle of Scanner

The proposed resonant fiber optic scanner mainly utilizes the inverse piezoelectric effect of CMPC and the resonance amplification effect of the transverse vibration of cantilever beam.

Figure 1c shows the structure diagram of the CMPCs. Each layer of the piezoelectric ceramic sheet is mechanically connected and electrically insulated, and external electrodes electrically connected to the internal electrodes. With a multilayer piezoelectric thin-layer structure, CMPC has a higher piezoelectric coefficient compared with the general piezoelectric material and can provide the same initial displacement at a lower voltage, and the scanner scan range is effectively amplified after resonant amplification. The preparation process generally involves preparing the piezoelectric ceramic paste into a thin film blank through a casting process, then coating the metal inner electrode on the thin film blank layer through a printing process, and then making a multilayer piezoelectric ceramic blank through a lamination, isostatic pressing and cutting process. Finally, a multilayer piezoelectric ceramic element with a mechanical structure in series and an electrical structure in parallel is formed by low-temperature co-fired process. CMPCs can superimpose the displacement produced by multiple single-layer piezoelectric ceramics to produce a larger displacement output. Therefore, CMPCs usually have a large piezoelectric coefficient d_33_. Most of the CMPCs used today are wafer-type, which further reduces the drive voltage, and have been widely used in piezoelectric actuators.

The excitation signals of two opposite CMPCs are arranged as in Figure 1b, and when one CMPC extends, the opposite CMPC shortens. The same is true for another group of CMPCs. By applying a sinusoidal voltage signal to one of the groups of CMPCs, the base of the cantilever capillary tube undergoes micron-scale displacement. Near the first-order resonance, the base displacement is amplified to the millimeter scale of the capillary tube free end displacement. By applying two drive signals A and B to the two groups of CMPCs in the horizontal and vertical directions, the displacement response of the free end of the capillary tube will form an X–Y plane scan pattern. Common space-filling 2D scanning includes spiral scans, Lissajous scans, propeller scans, etc. The different space-filling 2D scans are realized by specific drive signals, and the drive signals and the corresponding scan patterns are shown in Figure 2.

As shown in Figure 2a, the assumed X–Y scan pattern forms an Archimedean spiral scan when the drive signals are two sinusoidal waves with a phase difference of 90° amplitude modulated by a triangle wave. The frequency of the sine wave is near the natural frequency of the first-order transverse vibration of the capillary tube, while the frequency of the triangle wave is the frame rate of the scan. As a space-filling efficient 2D scanning, the spiral scanning pattern is the most used scan pattern, and its sampling rate can be adjusted to increase image resolution. The fiber’s tip position is also symmetric about the scan axis easing lens design.

As shown in Figure 2b, the assumed X–Y scan pattern is a Lissajous pattern when the drive signals in the two directions are sinusoidal waves with different frequencies but both near the natural frequency of the first order transverse vibration of the capillary tube. Lissajous scan is a highly redundant scan as the scan point passes through the same position several times in a single scan frame.

As shown in Figure 2c, the assumed X–Y scan pattern is a propeller pattern when the drive signals are two amplitude-modulated sine waves of the same frequency that are in phase and the modulated signals are sine and cosine waves, respectively. The propeller scan is slightly less inefficient, crossing the center of the scan every scan line.

When the scanner is operating in resonant mode, the tip displacement of is amplified and a large scanning area can be obtained. Analyzing the dynamics model of the scanner can provide a theoretical basis for the design of a fiber optic scanner.

## 3. Dynamics Model and Solutions

### 3.1. Linear Model of Free Vibration of Cantilever Beam

The proposed fiber optic scanner mainly uses the transverse vibration modes of the cantilever beam for scanning. In order to obtain better vibration performance, a high stiffness and easily machinable stainless-steel tube was chosen for the capillary tube. We consider the stainless-steel tube as a base-excited cantilever beam of transverse vibration. Therefore, the vibration model of the cantilever beam can describe the dynamical behavior. The structure of the base, connector and CMPC combination is sufficiently stiff and the mass of the stainless-steel tube is sufficiently small that we can consider the base to be fixed when not excited. Take $w\left(x,t\right)=w$ as the transverse vibration displacement of the beam, $\theta \left(x,t\right)=\theta$ as the angle of rotation along the neutral axis, $Q\left(x,t\right)=Q$ as the transverse shear (internal) force, $M\left(x,t\right)=M$ as the (internal) bending moment, $f\left(x,t\right)=f$ as the transverse external force acting per unit length of the beam and $I\left(x\right)=I$ as the cross-sectional moment of inertia of the beam section against its neutral axis. Figure 3a,b, shows the force analysis of a beam and its micro-element for transverse vibration. Applying Newton’s law to the micro-element in the transverse (z-direction) and x-direction, respectively, without considering the damping of the beam, gives
(1)$$\rho Sdx\frac{\partial^{2}w}{\partial t^{2}}=Q-Q-\frac{\partial Q}{\partial x}dx+fdx,$$
(2)$$\rho Idx\frac{\partial^{2}\theta }{\partial t^{2}}=-M+M+\frac{\partial M}{\partial x}dx-Qdx$$
where ρ is the density of the beam and S is the cross-sectional area of the beam.

According to the mechanics of materials [27], the relationship between Q,w,M,θ is as follows:(3)$$\frac{\partial \theta }{\partial x}=\frac{M}{EI},\;\theta -\frac{Q}{aSG}=\frac{\partial w}{\partial x},$$
where *a* is the shear factor, *E* is the material elasticity modulus and *G* is the material shear modulus. $\rho I\frac{\partial^{2}\theta }{\partial t^{2}}$ and $\frac{Q}{aSG}$ are usually second-order deviations. The former is the moment of inertia introduced by Rayleigh and the latter is the shear deformation introduced by Timoshenko. For Timoshenko beam, the effects of both are considered, but for Bernoulli-Euler beams, neither is considered [28]. Below, the equation of motion of Bernoulli–Euler beam is derived by ignoring these two second-order deviations, and then the partial differential equation for transverse free vibration of the beam can be written as [29]
(4)$$EI\frac{\partial^{4}w}{\partial x^{4}}+\rho S\frac{\partial^{2}w}{\partial t^{2}}=0$$

Take vibration displacement function:(5)$$w\left(x,t\right)=\phi \left(x\right)q\left(t\right)$$

By substituting (5) into (4) and combining the variable separation method, we obtain
(6)$$\frac{d^{2}q\left(t\right)}{dt^{2}}+w^{2}q\left(t\right)=0,$$
(7)$$\frac{d^{4}\phi \left(x\right)}{dx^{4}}-\frac{\rho S}{EI}w^{2}\phi \left(x\right)=0,$$

In our study, the beam of length *l* is a cantilever beam, whose boundary conditions can be described as
(8)$$x=0,\;\phi \left(0\right)=\phi^{\prime }\left(0\right)=0,$$
(9)$$x=l,\;\phi^{\prime \prime }\left(l\right)=\phi^{\prime \prime \prime }\left(l\right)=0,$$

Combined with Equations (8) and (9), the *n* order natural frequency and corresponding formation function of the beam with uniform and equal section can be obtained as follows:(10)$$\omega_{n}=\frac{X_{n}^{2}}{l^{2}}\sqrt{\frac{EI}{\rho S}}\left(X_{1}^{2}=3.516,\;X_{2}^{2}=22.03,\;X_{3}^{2}=61.69,\;\cdots \right),$$
(11)$$\phi_{n}\left(x\right)=\cos h\frac{X_{n}}{l}x-\cos\frac{X_{n}}{l}x-\xi_{n}\left(\sin h\frac{X_{n}}{l}x-\sin\frac{X_{n}}{l}x\right),$$

Among them,
(12)$$\xi_{n}=\frac{\sin hX_{n}-\sin X_{n}}{\cos hX_{n}+\cos X_{n}},$$

Equation (10) it can be used to determine the length of the stainless-steel tube to achieve the desired natural frequency. Meanwhile, the Equation (11) can be used to determine the tip position and angle under different amplitudes, which is an important basis for optimizing scanning. In practice, the cantilever beam is under forced vibration, so it is necessary to analyze the transverse forcing of the cantilever beam.

**Figure 3 micromachines-14-00517-f003:**
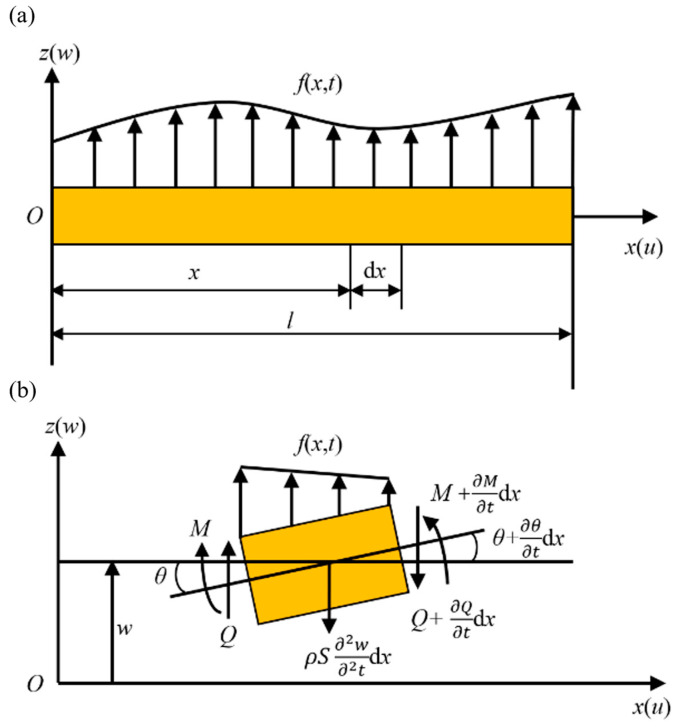
Diagram of force analysis of cantilever beam. (**a**) Force analysis of beam transverse vibration. (**b**) Force analysis of a micro-element on the transverse vibrating beam.

### 3.2. Linear Model of Free Vibration of Cantilever Beam with Damping

The Kelvin–Voight model can be used to describe the small amplitude strain of solid under periodic load [30]. Therefore, we used the Kelvin–Voight model to describe the effect of viscoelastic damping. It is assumed that the stress acting on the beam is proportional to the strain and strain rate. We can obtain [31]
(13)$$\sigma =Ew+\gamma \frac{dw}{dt}=E\left(w+\alpha \frac{dw}{dt}\right)$$
where σ is the stress, γ is the internal frictional coefficient and α is the intrinsic loss factor. For the external loss, the damping force can be assumed to be proportional to the cross section and the velocity. Then, the partial differential equation of the cantilever beam can be obtained:(14)$$\rho S\frac{\partial^{2}w}{\partial t^{2}}+EI\frac{\partial^{4}}{\partial x^{4}}\left(w+\alpha \frac{\partial w}{\partial t}\right)=0$$

Similarly, we assume that the response of the cantilever beam is
(15)$$w\left(x,t\right)=\phi \left(x\right)q\left(t\right)$$

Substitute (14) into (15) to obtain
(16)$$\rho S\frac{\partial^{2}}{\partial t^{2}}\phi \left(x\right)q\left(t\right)+EI\frac{\partial^{4}}{\partial x^{4}}\left[\phi \left(x\right)q\left(t\right)+\alpha \frac{\partial }{\partial t}\phi \left(x\right)q\left(t\right)\right]=0$$

That is,
(17)$$\rho S\phi \left(x\right)\frac{\partial^{2}q\left(t\right)}{\partial t^{2}}+EI\left[q\left(t\right)+\alpha \frac{\partial }{\partial t}q\left(t\right)\right]\frac{\partial^{4}\phi \left(x\right)}{\partial x^{4}}=0$$

Equation (17) is the dynamics of the cantilever beam under the Kelvin–Voight model. In order to solve the space–time solution of linear dynamic system with damping, divide the Equation (17) by $\left[q\left(t\right)+\alpha \frac{\partial }{\partial t}q\left(t\right)\right]\phi \left(x\right)$ and then move the second term to the right side to obtain
(18)$$\frac{\rho S}{\left[q\left(t\right)+\alpha \frac{\partial }{\partial t}q\left(t\right)\right]}\frac{\partial^{2}}{\partial t^{2}}q\left(t\right)=-\frac{EI}{\phi \left(x\right)}\frac{\partial^{4}\phi \left(x\right)}{\partial x^{4}}$$

Since the left side of Equation (18) is a function of time *t*, while the right side is a function of space position *x*, and they both equal a constant. Thus, it is possible to:(19)$$\frac{\rho S}{EI\left[q\left(t\right)+\alpha \frac{\partial }{\partial t}q\left(t\right)\right]}\frac{\partial^{2}}{\partial t^{2}}q\left(t\right)=-\frac{\frac{\partial^{4}}{\partial x^{4}}\phi \left(x\right)}{\phi \left(x\right)}=-\beta^{4}$$

It can be seen from Equation (19) that
(20)$$\frac{\partial^{2}}{\partial t^{2}}q\left(t\right)=-\beta^{4}\frac{EI}{\rho S}\left[q\left(t\right)+\alpha \frac{\partial }{\partial t}q\left(t\right)\right]$$

Moving the term from the right side of Equation (20) to the left side and expanding it to obtain
(21)$$\frac{\partial^{2}}{\partial t^{2}}q\left(t\right)+\frac{EI}{\rho S}\beta^{4}\alpha \frac{\partial }{\partial t}q\left(t\right)+\frac{EI}{\rho S}\beta^{4}q\left(t\right)=0$$

Let $\Omega =\sqrt{\frac{EI}{\rho S}}\beta^{2},\zeta =\sqrt{\frac{EI}{\rho S}}\frac{\alpha \beta^{2}}{2}$, then Equation (21) can be rewritten as follows:(22)$$\frac{\partial^{2}}{\partial t^{2}}q\left(t\right)+2\zeta \Omega \frac{\partial }{\partial t}q\left(t\right)+\Omega^{2}q\left(t\right)=0$$

Equation (22) is a second-order differential equation of the system time response. We find that adding viscoelastic damping to a cantilever beam is adding a viscoelastic damping term to the second-order differential equation describing the dynamics of the system. Equation (22) is a mass-spring-damping system with a single degree of freedom. According to the theory of ordinary differential equations, the solution to this differential equation has the following form:(23)$$q\left(t\right)=\bar{A}e^{i\Omega t}$$

Equation (23) is the time-free response of the system. In order to obtain the spatial solution, the new hypothesized solution (23) is substituted into the dynamical Equation (17) of the system to obtain
(24)$$\rho S\phi \left(x\right)\frac{\partial^{2}\bar{A}e^{i\Omega t}}{\partial t^{2}}+EI\frac{\partial^{4}\phi \left(x\right)}{\partial x^{4}}\left(\bar{A}e^{i\Omega t}+\alpha \frac{d}{dt}\bar{A}e^{i\Omega t}\right)=0$$

The simplification is as follows:(25)$$-\rho S\phi \left(x\right)\Omega^{2}+EI\frac{\partial^{4}\phi \left(x\right)}{\partial x^{4}}\left(1+i\alpha \Omega \right)=0$$

Divide both sides of Equation (25) by $EI\left(1+i\alpha \Omega \right)$ to obtain
(26)$$\frac{\partial^{4}\phi \left(x\right)}{\partial x^{4}}-\frac{\rho S\Omega^{2}}{EI\left(1+i\alpha \Omega \right)}\phi \left(x\right)=0$$

Let $\beta^{4}=\frac{\rho S\Omega^{2}}{EI\left(1+i\alpha \Omega \right)}$, then Equation (26) can be rewritten as
(27)$$\frac{\partial^{4}\phi \left(x\right)}{\partial x^{4}}-\beta^{4}\phi \left(x\right)=0$$

Equation (27) can be similar to Equation (19), thus
(28)$$\frac{\rho S}{EI\left[q\left(t\right)+\alpha \frac{\partial }{\partial t}q\left(t\right)\right]}\frac{\partial^{2}}{\partial t^{2}}q\left(t\right)=-\frac{\frac{\partial^{4}}{\partial x^{4}}\phi \left(x\right)}{\phi \left(x\right)}=-\beta^{4}$$

Equation (27) shows that the constant $\beta^{4}=\frac{\rho S\Omega^{2}}{EI\left(1+i\alpha \Omega \right)}$. This means that, given an arbitrary base motion amplitude excitation, we can adjust the value of the tip amplitude to obtain the correct tip amplitude. At the same time, the parameters of the stainless-steel tube can be designed according to the different scanning frequency requirements. Taking 200 Hz as an example, the relevant parameters of the stainless-steel tube can be substituted into the equation to plot the first three modes of the transverse vibration displacement response curve of the cantilever beam as shown in Figure 4. The stainless-steel tube has an inner and outer diameter of 0.5 mm and 0.7 mm, respectively, a density of 7.93 × 10^3^ kg/m^3^ and a modulus of elasticity of 200 GPa [32]. The dimensional parameters of the stainless-steel tube are shown in Figure 5, and the material parameters of the stainless-steel tube are shown in Table 1.

Figure 4 shows that when the scanning frequency is constant, the first-order transverse vibration requires the shortest length and the smallest excitation signal voltage to achieve the same transverse displacement. Therefore, we chose the first order transverse vibration as the operating mode of the scanner. The scanning frequency was set to 200 Hz, and the length of the cantilever was calculated to be approximately 55 mm.

## 4. Finite Element Simulation of Scanning System

### 4.1. Finite Element Simulation of Scanner

As the proposed resonant optical scanner utilizes the first-order transverse vibration mode of a cantilever beam, the dynamics analysis can guide the dynamic design of the scanner structure well. The scanner was simulated using the finite element simulation software ANSYS for dynamic simulation (ANSYS 19.2, ANSYS Inc., Houston, TX, USA). The co-fired multilayer piezoelectric (PZT-5H) ceramic AE1.65 × 1.65 × 5DF (Tokin Corp., Tokyo, Japan) was chosen, with the specific material parameters shown in Table 1. The height of the connector was chosen to be 3 mm in order to facilitate the drilling of the holes according to the scanner shown in Figure 5. QSn 6.5-0.1 (Tianjin Yitejia Steel Sales Co., Ltd., Tianjin, China) was chosen as the material for the base and connector with outstanding mechanical properties, and fixed constraints were applied to the base. In order to facilitate the clamping of the fixed base during the experiments, the base was designed to be square, without affecting the dynamics of the overall structure. Considering the small size and mass of the fiber optic, its effect on the vibration can be ignored and therefore the fiber optic is not included in the finite element simulation model. Finally, the corresponding finite element model is built according to the geometry in Figure 5.

### 4.2. Modal Analysis

Modal analysis is the basis of dynamics analysis and can determine the natural frequencies and vibration modes of each order of the scanning system. Using the finite element software ANSYS, a modal analysis of the scanning system shown in Figure 5 was carried out and the results are shown in Figure 6a,b.

The results of the modal analysis show that the natural frequencies of the two orthogonal modes of the first-order transverse resonance (mode A and B) *ƒ_A_* = 197.64 Hz, *ƒ_B_* = 197.87 Hz, with an error of only 1% from the theoretical results in Section 2. Combined with the theoretical analysis, it can be shown that the natural frequency of transverse vibration is mainly influenced by the length of the stainless-steel tube and cross-sectional moment of inertia, and is less influenced by the base, the connector and the CMPCs.

### 4.3. Transient Dynamic Analysis

Transient dynamics analysis is a method used to determine the dynamic response of a structure subjected to arbitrary and time-varying loads. The resonant fiber optic scanner is designed to use the superposition of the dynamic response of a structure under different alternating electrical excitations to form different space-filling 2D scanning patterns. Therefore, the transient dynamics analysis provides a good representation of the dynamic response characteristics of the scanner in different scanning patterns.

The scanner was analyzed for transient dynamics according to the different scanning patterns shown in Figure 2, with corresponding excitation voltages applied to each of the two groups of CMPCs for 300 cycles. Subsequently, a point on the end of the stainless-steel tube was selected to extract the displacement response of this point in the X–Y plane when the vibration reached steady state, as shown in Figure 7.

From the results of the transient dynamics analysis, the tip trajectory can fill the 2D space uniformly when it performs the spiral scanning as shown in Figure 7a. When the scanner performs a Lissajous scan as shown in Figure 7b, the shape of the scan trajectory shows a rectangle instead of a square shown in Figure 2b. This is mainly because the amplitudes of the cantilever beam are not exactly equal in the horizontal and vertical directions. When the scanner performs the propeller scan, shown in Figure 7c, the shape of the filled area is circular, but the scan trajectory does not pass through the center forming a circular ‘blind’ zone, unlike the X–Y assumed response in Figure 2c. This is mainly because assumed responses in Figure 2 are synthetic patterns of the drive signals and the actual responses of the scanner are not ideal sine signals or amplitude-modulated sine signals. When the scanning speed is fast, the scanner does not fully decay to rest during the previous cycle of vibration before proceeding to the next cycle. Therefore, the size of the “blind” zone can be reduced by decreasing the speed of the “blade” rotation. The simulation results showed that the designed structure can basically achieve three space-filling scanning patterns of spiral, Lissajous and propeller.

## 5. Experimental Testing

### 5.1. Scanner Dynamics Test 

In order to verify the accuracy of the proposed scanner dynamics theory and simulation calculations, a schematic prototype was machined and fabricated, and dynamics tests were carried out as well. The dynamics testing of the scanner enables the sweep (amplitude–frequency characteristics) curve of the scanner and the vibration modes at the corresponding frequencies to be obtained. The vibration response characteristics of the scanner were obtained using a scanning laser Doppler vibrometer (PSV-500-3D, Polytec Inc., Waldbronn, Germany) as shown in Figure 8.

First, the laser Doppler vibrometer (LDV) host generates a signal, which is then passed through a power amplifier (TD250, PiezoDrive Pty Ltd., Callaghan, NSW, Australia) to excite the scanner prototype to produce the mechanical vibration response. The vibration response is captured by the sensor of the laser vibrometer to the host computer. The data are then processed by professional software to obtain the corresponding amplitude and frequency response curves as well as the vibration patterns at the corresponding frequencies.

The experimental results showed that the resonant frequencies of the two orthogonal first-order transverse modes of the optical scanner were both 200 Hz, indicating that the natural frequencies obtained from the theoretical and simulation calculations of the scanner were consistent with the experimental tests, and a comparison of them is shown in Table 2. The main reasons for the inconsistency between the measured frequencies and the results obtained from the theoretical and Finite Element Analysis (FEA) calculations are (i) the influence of the epoxy resin adhesive is ignored in the FEA; (ii) the effect of the fiber optic was ignored in the FEA; and (iii) the actual machining and assembly process will produce a slight error compared to the finite element model. The percentage error between the actual test results and the theoretical calculation results was calculated by dividing the absolute value of the difference between the theoretical value and the test value by the theoretical value and multiplying it by 100%. Additionally, the percentage error between the actual test results and the finite element simulation analysis was calculated by dividing the absolute value of the difference between the FEA value and the test value by the FEA value and multiplying it by 100%. The former was within 1% and the latter was within 1.5%, which is in line with the expected performance.

### 5.2. Laser Scanning Experiments

In order to further verify the feasibility of the proposed scanner and the correctness of the theoretical and simulation results, a resonant fiber optic scanner prototype shown in Figure 9 was built and two groups of CMPCs were excited at 50 V_p-p_ and 200 Hz AC voltages to obtain the scanning trajectory shown in Figure 10a and Figure 10b, respectively. When the two groups of CMPCs are excited simultaneously and the phase difference between the excitation voltages is π/2, the trajectory shown in Figure 10c can be obtained. When the drive voltage of one of the phases is reduced, the scan trajectory shown in Figure 10d is obtained.

From Figure 10a,b, the scanning trajectories in the horizontal X and vertical Y directions in the resonant state are two relatively stable line segments on the X and Y axes. From Figure 10c,d, the amplitudes of the horizontal X and vertical Y directions are close to each other in the resonance state, which can form a circular trajectory. The circularity of the trajectory of the circular scanning was calculated to be 0.9, indicating that the scan trajectory is close to a perfect circle. At the same time, by adjusting the voltage peak-to-peak value in the X direction, it does not affect the vertical direction displacement, and an elliptical trajectory is obtained. Scan trajectories were recorded for different drive methods of the scanner, such as circular scanning, Lissajous scanning, spiral scanning and propeller scanning, as can be seen in the Appendix A. The abovementioned results show that the basic scanning characteristics of the scanner are superior.

To further illustrate the feasibility of the scanner for space-filling scanning, the actual displacement response of the free end of the stainless-steel tube was tested using a Position Sensitive Detector (PSD, DL100-7, First Sensor, Berlin, Germany), and the diagram of the measuring system is shown in Figure 11. First, a function generator (AFG1062, Tektronix Inc., Beaverton, OR, USA) generated the excitation waveform, which was applied to the four CMPCs through the power amplifier TD250 (PiezoDrive Pty Ltd., Callaghan, NSW, Australia), and the stainless-steel tube resonated, driving the fiber optic to scan. The light source is connected to one end of a fiber optic (Beijing Pi-Optics Co., Ltd., Beijing, China) and the laser is conducted to the other end of the fiber, where it casts a spot on the surface of the PSD chip in front of it. At this point, the PSD senses the change in position of the spot and generates an X/Y directional position signal which is sent to a 14-bit data acquisition card (PXI8502S, Art Technology Co., Ltd., Beijing, China). Finally, the steady-state data are processed and synthesized by a PC host computer to obtain the spot motion trajectory, as shown in Figure 12. The PSD sensor is a square active area position sensing p–i–n photodiode with 10 mm × 10 mm active area. The position detection error of the PSD was measured to be ±1%. As the range of the 2D PSD was only 10 mm × 10 mm, the scanning range of the scanner was adjusted to be within the range of the PSD by adjusting the excitation voltage. The positioning accuracy of the measurements was tested to be 5 µm in both the x and y directions.

As can be seen from the spiral scanning results in Figure 12a, the scan trajectory efficiently and equally spaced filled the 2D circular space. However, there was a certain skew in the spiral scanning in the experiment. This is since the scanner was not perfectly centrosymmetric due to machining and assembly errors. For example, the dynamics test results shown in Figure 13 also show that the vibration performance in the X and Y directions was not the same. This small difference is particularly magnified in the resonant state. In this case, the process has been optimized to minimize the machining and assembly errors of the device. In addition, when applying the drive signal, the frequency of the signal can be shifted slightly away from the resonant frequency to reduce this effect.

As can be seen from the Lissajous scan results in Figure 12b, the actual test scan trajectory of the Lissajous scan constitutes a parallelogram-like region rather than a symmetrical pattern, which is mainly caused by errors in the machining assembly. Furthermore, the experimentally measured scanning area was not rectangular, but it can be found that the scanning range in the X direction was significantly smaller than in the Y direction, which corresponds to simulation analysis.

As can be seen from the propeller scan results in Figure 12c, the short axis of the elliptical trajectory became progressively smaller as the angle of rotation changed, with the scan trajectory becoming a straight line at 45°, 135°, 225° and 315°. This is due to the more pronounced vibration coupling in both the horizontal and vertical directions as the rotation reaches these angles. In addition, the graph shows that the three trajectories pass well through the same position after several cycles, which proves the good repeatability of the scanner.

We also tested the scanner’s performance in air and in deionized water. The specific comparison results are shown in Table 3. Test results showed that CMPCs-driven scanner requires a lower voltage and smaller size, especially in the axial dimension, to achieve comparable performance to the piezo tube-driven structure. The size of the piezo tube as actuator in Reference [4] is Φ7 × 35 mm; the size of the CMPCs parts in this work is only 5 mm × 5 mm × 13 mm, which is 22 mm shorter in axial dimensions in comparison. As the natural frequency increased, the vibration velocity increased and was subjected to increased damping force, which led to a decrease in amplitude. Additionally, the CMPCs-driven scanner, compared with the piezo tube-driven scanner, could achieve 67% and 60% of the scanning range in air and water, respectively, while having a higher working frequency and only 1/10 of the driving voltage, which is sufficient to illustrate the superior performance of the designed scanner. The test results also imply a broader future for the CMPCs-driven scanners in terms of interventional diagnostics, scanning, therapy and so on.

## 6. Conclusions

In this paper, a resonant fiber optic scanner driven by four CMPCs was first successfully designed and prepared. By means of a clever arrangement of four CMPCs, two homogeneous orthogonal first-order transverse vibration modes of the cantilever beam structure were excited respectively. By applying different excitation signals, linear, circular, and three space-filling scanning patterns could be realized. The experimental results showed that under the condition of excitation voltage of 50 V_p-p_ and excitation frequency of 200 Hz, the scanner could achieve the linear scanning range of 10 mm in X and Y directions, as well as spiral and propeller scanning with a diameter 10 mm and Lissajous scanning with a length and width of 10 mm. In particular, the scanner driven by four CMPCs requires a lower excitation voltage and a more compact structure to achieve the same performance as piezo tube-driven structures, especially in terms of axial dimensions of the devices. The designed resonant fiber optic scanner features a compact structure, small size, good scanning repeatability, low driving voltage and high energy conversion efficiency. These advantages make it promising for projection, processing, illumination, image signal acquisition and therapy in the fields of optical precision instruments and clinical endoscopes.

## Figures and Tables

**Figure 1 micromachines-14-00517-f001:**
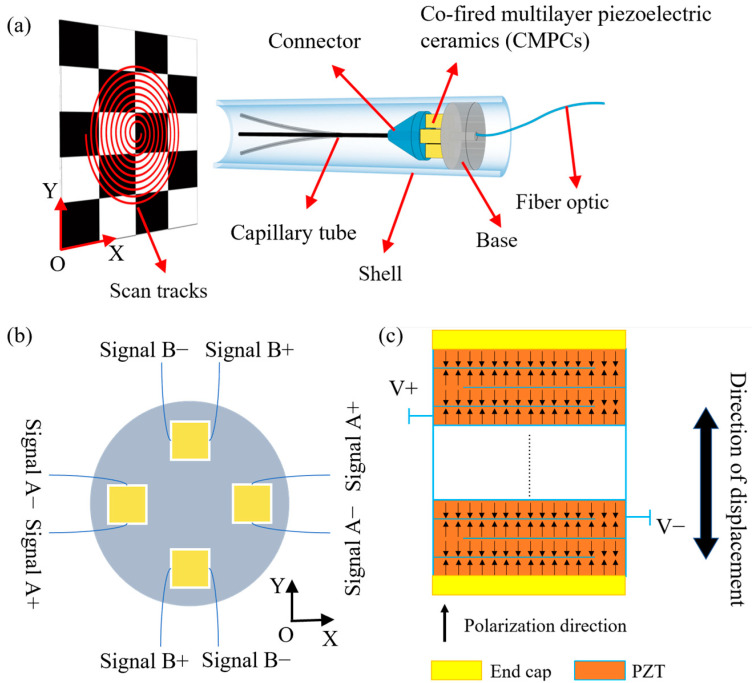
Schematic design of the scanner. (**a**) Structure diagram of the scanner. (**b**) Distribution of the four CMPCs (yellow parts) and voltage excitation strategy for them. (**c**) Structure diagram of the CMPC.

**Figure 2 micromachines-14-00517-f002:**
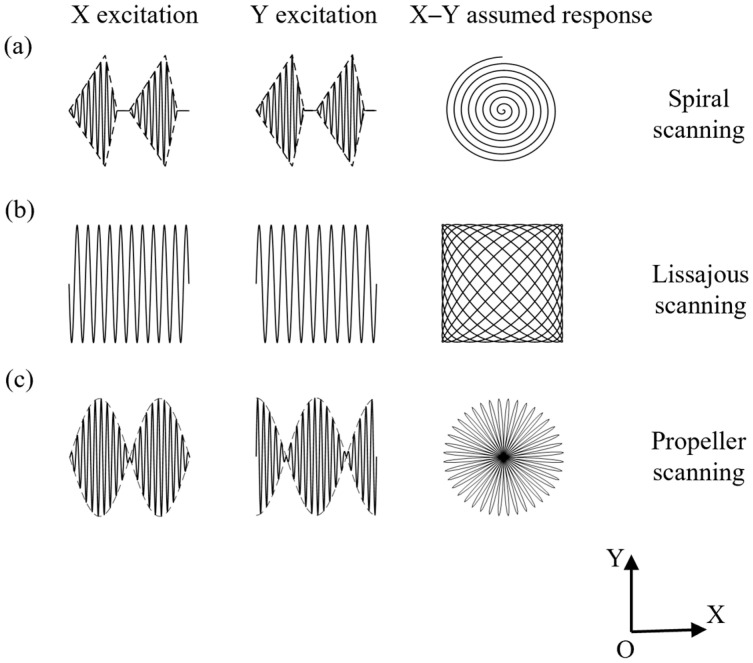
Different scanning patterns of resonant scanners: (**a**) spiral scanning; (**b**) Lissajous scanning; (**c**) propeller scanning.

**Figure 4 micromachines-14-00517-f004:**
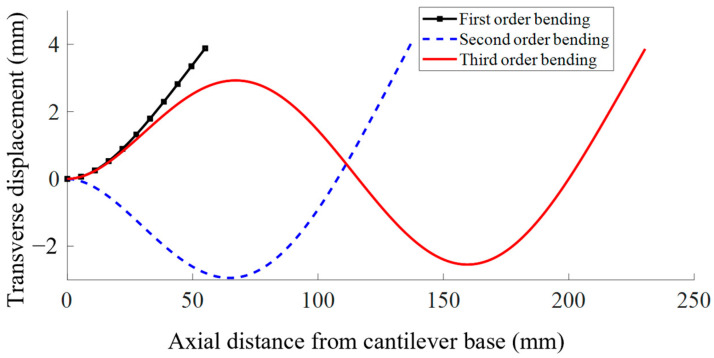
First three orders of transverse vibration displacement response curves.

**Figure 5 micromachines-14-00517-f005:**
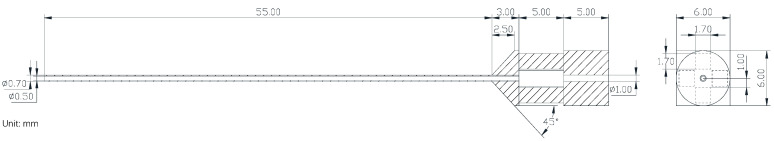
Schematic diagram of the scanner geometry.

**Figure 6 micromachines-14-00517-f006:**
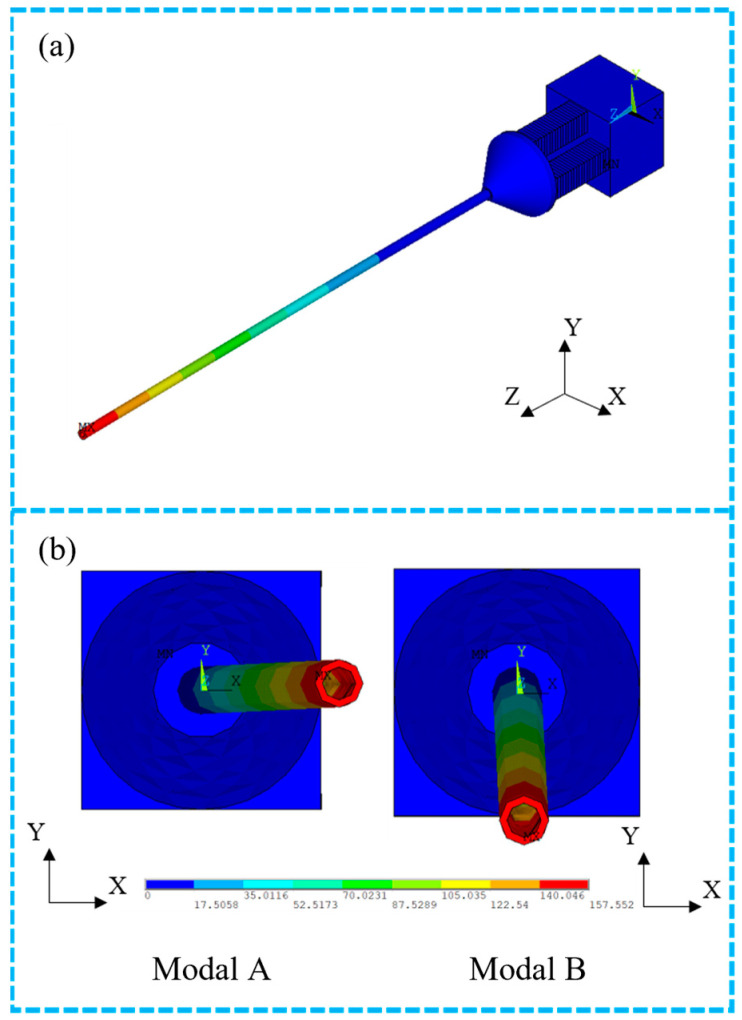
Modal analysis results of the scanner. (**a**) Bending vibration of the cantilever beam. (**b**) Two orthogonal modals (A and B) calculated by modal analysis.

**Figure 7 micromachines-14-00517-f007:**
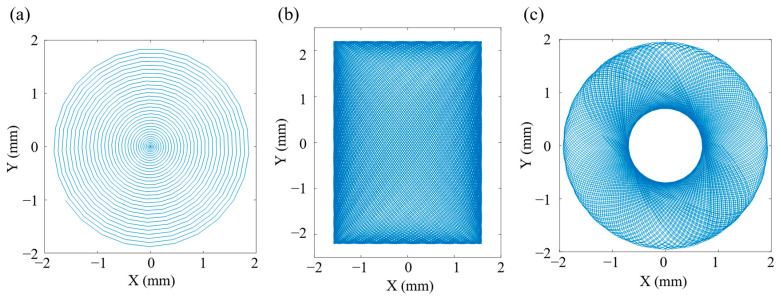
Transient analysis results of scanner: (**a**) spiral scanning pattern; (**b**) Lissajous scanning pattern; (**c**) propeller scanning pattern.

**Figure 8 micromachines-14-00517-f008:**
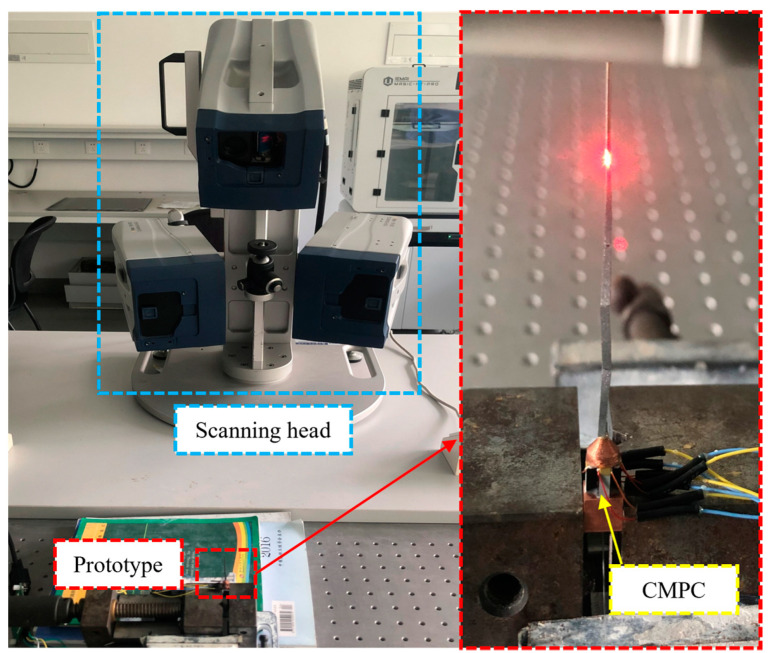
3D vibration measuring platform.

**Figure 9 micromachines-14-00517-f009:**
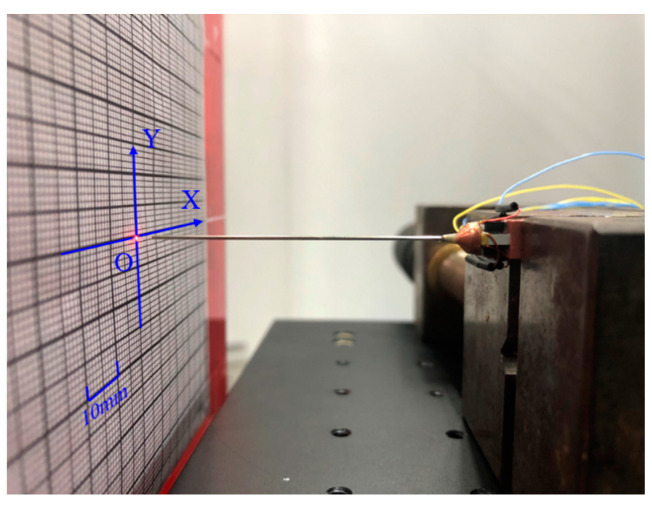
Principle prototype of optic scanner driven by CMPCs.

**Figure 10 micromachines-14-00517-f010:**
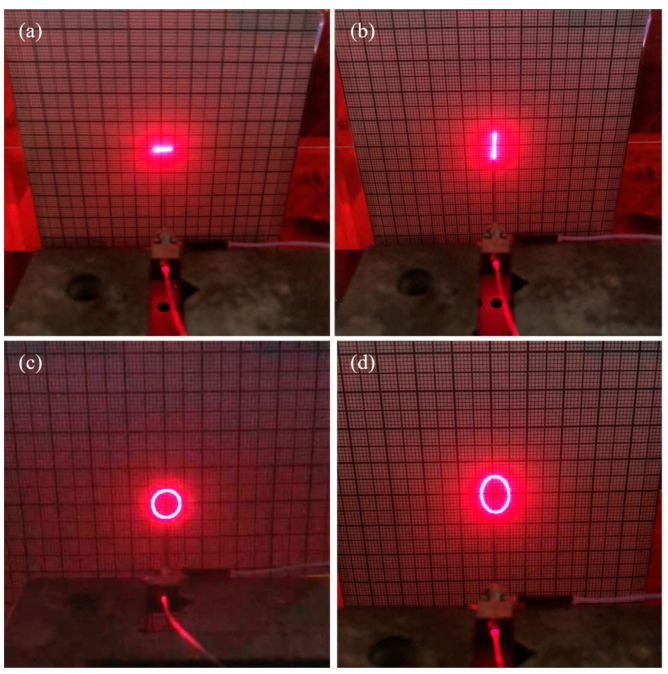
Photos of four types of simple scans. (**a**) Linear scanning in X direction. (**b**) Linear scanning in Y direction. (**c**) Circular scanning. (**d**) Elliptical scanning.

**Figure 11 micromachines-14-00517-f011:**
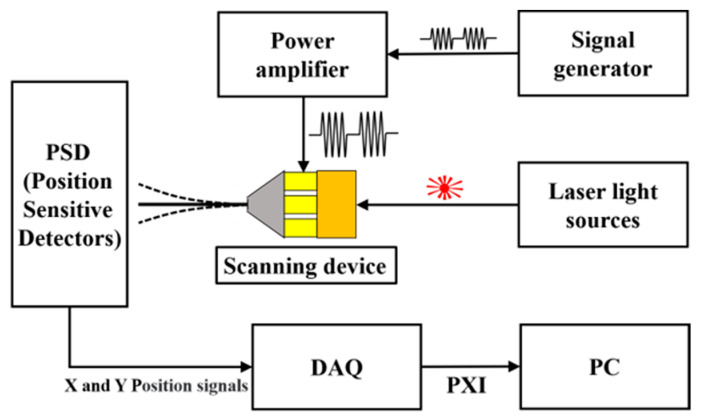
PSD displacement response test schematic.

**Figure 12 micromachines-14-00517-f012:**
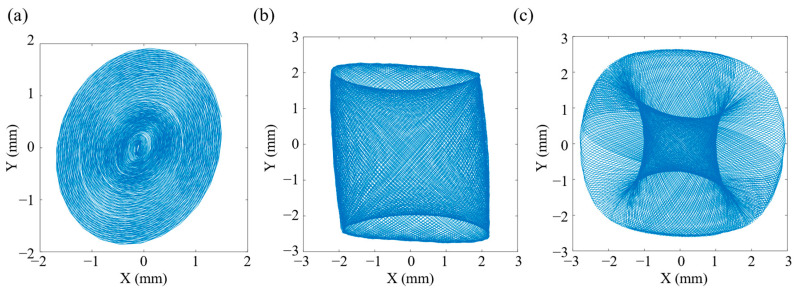
Measurement results of different scanning patterns by PSD. (**a**) Spiral scanning, (**b**) Lissajous scanning and (**c**) propeller scanning.

**Figure 13 micromachines-14-00517-f013:**
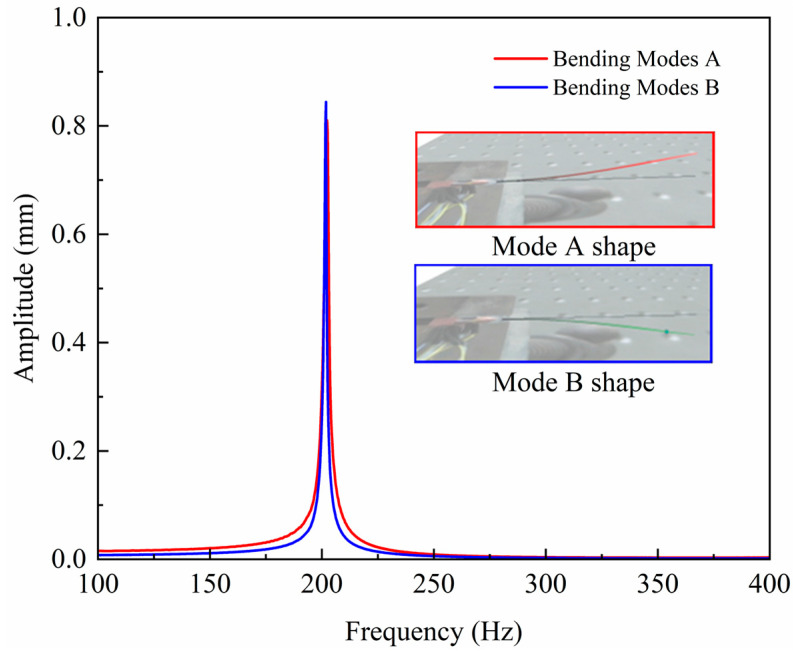
3D vibration measurement results.

**Table 1 micromachines-14-00517-t001:** Material parameters of the scanner.

Material	QSN	Stainless Steel	PZT-5H
Young’s modulus (Gpa)	110	200	$\left[\begin{array}{cccccc} 127.0 & 80.2 & 84.7 & 0 & 0 & 0\\ 80.2 & 127.0 & 84.7 & 0 & 0 & 0\\ 84.7 & 84.7 & 117.4 & 0 & 0 & 0\\ 0 & 0 & 0 & 23.5 & 0 & 0\\ 0 & 0 & 0 & 0 & 23.0 & 0\\ 0 & 0 & 0 & 0 & 0 & 23.0 \end{array}\right]$
Piezoelectric matrix (C/N)	/	/	$\left[\begin{array}{ccc} 0 & 0 & -6.6\\ 0 & 0 & -6.6\\ 0 & 0 & 23.2\\ 0 & 0 & 0\\ 0 & 17 & 0\\ 17 & 0 & 0 \end{array}\right]$
Poisson’ ratio	0.34	0.3	/
Density (kg/m^3^)	8.78 × 10^3^	7.93 × 10^3^	7.50 × 10^3^

**Table 2 micromachines-14-00517-t002:** First-order natural frequencies obtained from theory, FEA and experiment.

Modal	Theoretically Calculated (Hz)	FEA Calculated (Hz)	Experimentally Measured (Hz)
Modal A	199.79	197.64	200
Modal B	199.79	197.87	200

**Table 3 micromachines-14-00517-t003:** Performance comparison with the piezo tube scanner.

Parameters	Piezo Tube Scanner [4]	This Prototype
Piezoelectric actuator size (mm)	Φ7 × 35	5 × 5 × 13
Driving voltage (V_p-p_)	500	50
Working frequency (Hz)	112	200
Amplitude in air (mm)	15	10
Amplitude in water (mm)	2	1.2

## Data Availability

The data presented in this study are available on request from the corresponding author.

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
