# Peer review of "Structural Design and Experimental Studies of Resonant Fiber Optic Scanner Driven by Co-Fired Multilayer Piezoelectric Ceramics"

_micromachines, 2023, doi:10.3390/mi14030517_

Round 1

Reviewer 1 Report

Structural Design and Experimental Studies of Resonant Fiber 2 Optic Scanner Driven by Co-fired Multilayer Piezoelectric Ceramics.

In general, this article is pertinent to the Journal's approach. Authors are requested to consider to the following remarks:

1.  No mention is made of whether elements of originality are contributed. It is necessary to emphasize them where appropriate, including in conclusions.

2.       Identify improvements or elements of differentiation in the CMPC, if any. If applicable, add or emphasize them in the appropriate place.

3.       No scanned object is presented yet. It is highly desirable to add an example.

4.       In Figure 1, in (a) there is no label indicating the location of the CMPCs, it is suggested to add it for immediate identification.

5.       It is suggested to separate Figure 3, in (a) and (b) sections.

6.       On the properties of the stainless-steel tube, add the reference of its data (lines 264-266).

7.       In Figure 4, a separation is required between description labels and their units.

8.       In Figure 5, please add the units.

9.       In Table 1, the units are in a different system of units, please homologate.

10.   It is suggested to separate Figure 6 in (a) and (b) sections.

11.   In Figure 8, the prototype image is not clear, please replace it.

12.   In Table 3, the size of the piezo tube scanner seems wrong, please correct it. And, please added to its description the advantages or disadvantages of the working frequency values, as well of the amplitude in air, and in water.

13.   In the conclusions, highlight the specific improvements achieved in the CMPCs, if any.

14.   In line 454, specify the reasons of the following affirmation: "... especially in axial dimensions".

15.   In the references, use the format provided in the corresponding template of the Journal. All of them need to be reviewed and adjusted.

Author Response

Dear Reviewer,

My revision and response are listed in the attachment. 

Kind Regards!

Reviewer 2 Report

Dear Authors,

My comments and review points are listed in the attachment. 

Kind Regards

Author Response

Dear Reviewer,

Our revision and responses are listed in the attachment. 

Kind Regards!

Reviewer 3 Report

In my opinion, the manuscript is suitable for publication in Micromachines journal, but authors must complete a minor revision. The manuscript should be revised according to the following comments:

1. The chapter “Introduction” should be improved:
a) in line 76 the term: "d31 vibration mode" is used. d31 coefficient is not vibration mode,
b) in lines 86-87.86-87 Authors write: "d33 coefficient of up to tens of thousands". It is not known what this sentence means,
c) in lines 96-98 the authors indicate what their article is about. However, it is not known what is new in their article compared to the state of the art, for example, compared to [23]. The authors must clearly state what is new in their article in a scientific sense.

2. The chapter “Structure Design and Working Principle” must be improved:
a) coordinate systems should be added in Figures 1a, 1b, 1c,1c and 1d,
b) piezoelectric material parameters (d, s, etc.) should be given,

3. The chapters “Dynamics model and solutions” must be improved:
a) the methodology of the numerical experiment in which the runs presented in Fig. 4 were performed should be described before this figure,

4. The chapters “Experimental testing” must be improved:
a) the parameters of the sensors used should be given,
b) the diagram of the measuring system should be added,
c) the accuracy of the measurements performed should be specified,
d) in lines 364-367 the authors give the value of errors. It should be clearly explained how the given error values were calculated,
e) drawings should be added to compare the numerical and laboratory results.

Author Response

(The authors gave the same response as above.)

Round 2

Reviewer 1 Report

Dear Editors:

Thank you for inviting me to review the article "Structural Design and Experimental Studies of Resonant Fiber Optic Scanner Driven by Co-fired Multilayer Piezoelectric Ceramics", which fits the scope of the journal and presents interesting findings and design contributions. It certainly focuses on the design and performance testing of the scanner, so there is still work to be done in terms of its application, which could be published in future.

The authors have made the required corrections, and included the additional comments requested, the manuscript has improved markedly. I have no further suggestions on the form and content of this article.

Reviewer 2 Report

Dear Authors,

You have modified the paper according to my comments. The paper can be published after minor revision (methodological errors and text editing). 

Kind Regards